# Answering Schrödinger’s “What Is Life?”

**DOI:** 10.3390/e22080815

**Published:** 2020-07-25

**Authors:** Stuart Kauffman

**Affiliations:** Institute for Systems Biology, Seattle, WA 98109, USA; stukauffman@gmail.com

**Keywords:** aperiodic solid, code script, causally efficacious, boundary conditions, information, thermodynamic work, constrained release of energy, constraint closure, delayed production of entropy, open-ended evolution, entailing laws, Newtonian paradigm

## Abstract

In his “What Is Life?” Schrödinger poses three questions: (1) What is the source of order in organisms? (2) How do organisms remain ordered in the face of the Second Law of Thermodynamics? (3) Are new laws of physics required? He answers his first question with his famous “aperiodic solid”. He leaves his second and third questions unanswered. I try to show that his first answer is also the answer to his second question. Aperiodic solids such as protein enzymes are “boundary conditions” that constrain the release of energy into a few degrees of freedom in non-equilibrium processes such that thermodynamic work is done. This work propagates and builds structures and controls processes. These constitute his causally efficacious “code script” controlling development. The constrained release of energy also delays the production of entropy that can be exported from cells as it forms. Therefore, cells remain ordered. This answers his second question. However, “What is life?” must also ask about the diachronic evolution of life. Here, the surprising answer to this extended version of Schrödinger’s third question is that there *are* no new entailing laws of physics. No laws at all entail the evolution of ours or any biosphere.

## 1. Introduction

My purpose in this article is to discuss quantum physicist Erwin Schrödinger’s superb book, “What Is Life,” published in 1944 [1], the answers he gives and the very large problems with which he leaves us. 

In “What is Life” Schrödinger does *not* ask what “Life Is” or how life emerged from non-life then evolved.

Schrödinger asked three major questions: (1)What is the source of order in organisms?(2)How do organisms remain ordered in the face of the second law of thermodynamics?(3)Does life require new laws of physics?

I shall propose that Schrödinger’s answer to the first question, his famous aperiodic solid, is also the still unarticulated answer to his second question. The answer to his second question implies that new laws of physics are not needed to account for cells remaining ordered in face of the Second Law of Thermodynamics. This is his third question. However, an extended version of his third question asks whether new laws of physics are required to account for the evolution of the biosphere. The astonishing answer is that no entailing laws of physics exist. The biosphere blossoms in its wondrous diversity entailed by no laws.

## 2. Toward the Aperiodic Solid

Schrödinger notes that there is a source of “order from disorder” seen in statistical mechanics, for example the diffusion of a drop of milk in a coffee cup to a uniform mixture at equilibrium. Diffusion yields order from disorder. He notes that the basis of this macroscopic order is the chaos of the random motion of atoms. 

Then, Schrödinger notes that as the number of molecules in the system is reduced, the system becomes more random. Macroscopic order from disorder depends upon a large number of atoms.

He next considers recent evidence from the X-ray induction of heritable mutations that the hereditary material, genes, can have only on the order of a thousand atoms. Thus, “order from disorder” cannot explain the order in organisms. 

He concludes that the order in organisms must be due to quantum mechanically stable molecules, which are stable despite having only a small number of atoms. 

Schrödinger then brilliantly concludes that the solid will not be “periodic,” like quartz, for these are “dull.” Rather the hereditary material will be an aperiodic crystal that contains, in some kind of code-script, the entire pattern of the individual’s future development. The code script will be efficacious in bringing about the organism. 

“**In physics we have dealt hitherto only with *periodic crystals***. To a humble physicist’s mind, these are very interesting and complicated objects; they constitute one of the most fascinating and complex material structures by which inanimate nature puzzles his wits. Yet, compared with the aperiodic crystal, they are rather plain and dull. **The difference in structure is of the same kind as that between an ordinary wallpaper in which the same pattern is repeated again and again in regular periodicity and a masterpiece of embroidery, say a Raphael tapestry, which shows no dull repetition, but an elaborate, coherent, meaningful design traced by the great master.”**

“It is these chromosomes … that contains in some kind of code-script the entire pattern of the individual’s future development and of its functioning in the mature state. Every complete set of chromosomes contains the full code......”

It is Schrödinger’s brilliant intuition of an aperiodic solid that contains “in some kind of code-script the entire pattern” …that inspired Watson and Crick in their 1953 discovery of the structure of DNA, which Avery had shown to be the genetic material a few years earlier.

DNA is precisely an aperiodic solid. In the next decades, the existence of the genetic code was deduced and worked out.

Thus, Schrödinger concludes the order in organisms does not come by “order from disorder, as in diffusion, but Order from Order.” 

## 3. Schrödinger’s Second Question 

How do organisms remain ordered in face of the second law of thermodynamics?

“How would we express in terms of the statistical theory the marvelous faculty of a living organism, by which it delays the decay into thermodynamical equilibrium (death)?... the device by which an organism maintains itself stationary at a fairly high level of orderliness... really consists in continually sucking orderliness from its environment.”

Schrödinger’s phrase, sucking orderliness from its environment, is also called “negentropy” [1].

## 4. Schrödinger’s Third Question

Critically, Schrödinger does not see how his second question is answered by the known laws of classical and quantum physics:

“We must therefore not be discouraged by the difficulty of interpreting life by the ordinary laws of physics. For that is just what is to be expected from the knowledge we have gained of the structure of living matter. We must also be prepared to find a new type of physical law prevailing in it. Or are we to term it a non-physical, not to say a super-physical, law?”

## 5. Non-Equilibrium Dissipative Structures

Part of the answer to Schrödinger’s second question is already contained in his phrase, “really consists in continually sucking orderliness from its environment”. In systems displaced from equilibrium by the flow of matter and energy through them, macroscopic patterns can emerge. 

Dissipative Structures were unknown when Schrödinger wrote in 1944. 

Ilya Prigogine and his group led a large effort to study the onset of ordered patterns in far from equilibrium physical and chemical systems and termed such patterns “dissipative structures” [2]. Examples include whirlpools, tornados, hurricanes, Benard cells [3], and the Belosov Zhabatinski reaction [3]. The Giant Red Spot on Jupiter is an example [3].

In these systems a macroscopic pattern emerges as energy flows through the system that is thereby displaced from equilibrium by that energy flow. 

The Benard Cell is an instructive example. Here, there is a pan with a shallow layer of viscous liquid, for example, oil. The pan is heated slowly from below, creating a temperature gradient, hotter on the bottom than top of the fluid.

The temperature gradient induces an overall heat flow to the environment. When the gradient is small, the heat flow to the environment is achieved by conduction of random motions of the oil molecules. When the temperature gradient surpasses the Rayleigh threshold, convective cells arise and dissipate heat more effectively. The convective cells are the Benard cells.

Benard cell patterns are sustained by the continuous flow of energy through the system that results by heating the pan from below. The form of these cells depends upon the *boundary conditions* created by the shape of the pan. The Benard cells can be hexagonal or rolls. Critically, if the shape of the pan, i.e., the boundary conditions, are changed, so too are the shape and overall pattern of the Benard cells.

## 6. Towards an Answer to Schrödinger’s Second Question

I propose below that Schrödinger’s aperiodic solid that constitutes a “code-script” that is “effective for the formation of the organism” is because the aperiodic solid is one or a set of boundary conditions that thereby control the dynamics of the non-equilibrium system in living cells and organisms. Thus, as we will see in more detail below, Schrödinger’s answer to his first question *is* the answer to his second question.

## 7. Thermodynamic Work and Boundary Conditions: Beyond the Limitation of Dissipative Structures

Dissipative structures such as hurricanes, whirlpools and Benard cells [3], do not become very complex and increasingly diversified in their complexity.

By contrast, the evolution of life started with rather simple protocells, followed by single cell organisms such as Archea and bacteria, then eukaryotes, then multi-celled organisms. This was followed by the Cambrian Explosion 550 million years ago with the flowering of some 13 extant phyla to millions of distinct species today, bacteria to whales and redwood trees in an open-ended evolution.

What is the difference between dissipative structures and evolving life?

Part of the reason that dissipative structures do not become indefinitely complex is that they do *not* construct their own boundary conditions. Organisms, as we shall see, *do* construct their own boundary conditions and do this by carrying out thermodynamic work to construct the very same boundary [4].

## 8. Work, Constraints and the Work–Constraint Cycle

Work is force acting though a distance.

Work is more: Work is the constrained release of energy into a few degrees of freedom, Atkins [5].

An example is a cannon and cannon ball inside the cannon. The explosion of the powder releases energy into the few degrees of freedom left over as created by the cannon, which constrains the release of energy into those few degrees of freedom, so that the cannon ball can only move down the cannon shaft.

Thermodynamic *work* is done on the cannon ball. The powder explosion is exergonic, the resulting movement of the accelerated cannon ball is endergonic. When the power explodes, part of the released energy is dissipated as heat, producing entropy. But part of the released energy does thermodynamic work to propel the cannon ball.

Critically, were the cannon not present and the powder exploded on a flat plate next to the cannon ball, the ball would hardly move and most of the energy would be dissipated rapidly as heat, thereby rapidly producing entropy, and less work would be done. Thus, the constrained release of energy that does work in a linked exergonic–endergonic process produces *work* and at the same time, *delays* the production of Entropy *longer* than were the constraints not present. 

This point is fundamental: When *more* work is done in a *more* constrained release of energy in a non-equilibrium process, the production of entropy is *more delayed* in the process than were the process less constrained. This will be a central part of the answer to Schrödinger’s second question. 

## 9. Propagating Macroscopic Thermodynamic Work

In an automobile, the explosion of gas causes the piston to move, which causes the crankshaft to turn, which causes the wheels to turn. Thermodynamic work propagates in a coupled set of exergonic and endergonic processes from the gas explosion to the wheels turning. This is Propagating Work [6]. A large number of macroscopic changes happen in the world.

The cannon, which constitutes the constraint on the release of energy into a few degrees of freedom, is a boundary condition. Similarly, in an automobile, the cylinders and pistons, gears and escapements are boundary condition constraints on the release of energy from the exploding gas into a few degrees of freedom that constitutes propagating thermodynamic work, by which the engine leads to the physical motion of the car on the road.

## 10. The Work Constraint Cycle

Very shortly after the Big Bang, the universe was a hot quark gluon soup at thermodynamic equilibrium [7]. There were no cannons, rods and pistons. Where then, since the Big Bang, did the cannon or cylinder and piston boundary conditions come from? The obvious but surprising answer is that thermodynamic work was required to construct the cannon, the cylinders, the pistons, the gears and escapements that could then be used as boundary conditions.

We find ourselves in an unexpected place: Without constraints on the release of energy in non-equilibrium processes, there can be no work. But it typically takes work to construct constraints! This is the “work constraint” cycle: Without constraints, there is no work. Without work no constraints can be constructed [6]. 

In our human machines to date, *we* organize the work to construct structures, often low energy stable structures, such cannons and pistons that then are used as the boundary condition constraints on the release of energy into a few degrees of freedom such that propagating thermodynamic work is done. Then, we organize these boundary conditions linked to non-equilibrium processes such that our machines work. Engines turn over and cars move.

So again, since the Big Bang, where do boundary conditions come from? 

We will see below that living cells and organisms do thermodynamic work to construct their own boundary conditions, in what is called “Constraint Closure” [4,8]. Automobiles do not construct their own boundary conditions. 

## 11. In Classical Physics the Boundary Conditions Can Vary “Freely” So Can Carry a Causally Effective “Code Script”

Newton’s framework for classical physics is his law of universal gravitation and his three laws of motion that couple the variables in differential equation form. These laws of motion plus the initial and boundary conditions, which are *not* themselves specified in the law of motion, enable integration of the differential equations to yield the entailed trajectory of the system in its phase space. An example would be the billiard balls moving on a billiard table. 

The boundary conditions determine what the phase space of the system is. For example, on a billiard table, the boundary conditions given by the table boundaries thereby specify all possible positions and momenta of the balls on the table. This set of all possible positions and momenta constitutes the “phase space” of the system.

Keeping the same boundary conditions and altering the initial conditions alters the trajectories of the balls in the “same” phase space. Altering the boundary conditions, for example the shape of the billiard table, alters the very phase space of the system. But Newton’s laws leave “free” what the boundary conditions might be.

A simple example is a Pee Wee golf course I visited recently. It consisted of 18 “holes”. Each hole was created by arranging 6 × 6 inch and long timbers on the ground in various arrangements. For example, the major timbers were parallel to one another creating pathways, then turned corners at various angles. Short timbers were arranged jutting out into the pathways the parallel boards created. The golf ball, once hit, caromed off the long timbers and jutting short timbers in complex ways. The long and short timbers are boundary conditions.

If the short timbers are rearranged, the motion of the golf ball, after it is hit, will change. But “we” are “free” to rearrange the long and short timber boundary conditions in many ways. For such a timber arrangement to carry “information”, it is essential that the timbers could, counterfactually, have been located in a different way. This “counterfactual” captures the “we are ‘free’ to rearrange the short timbers”. Rearranging the timbers in some arbitrary way alters the phase space of the system in some arbitrary way.

## 12. Toward Schrödinger’s Second Question

As just noted, rearranging the timbers in some arbitrary way alters the *phase space* of the system in some arbitrary way. 

In evolving organisms, with DNA RNA and encoded protein synthesis, once synthesized, these molecules are all “aperiodic solids” that constitute boundary conditions. These boundary conditions constrain the release of energy in myriad enzyme catalyzed chemical reactions into a few degrees of freedom that does thermodynamic work to construct further and often changing aperiodic enzymes that are further boundary conditions.

In cells, thermodynamic work is done to synthesize RNA and proteins. The new RNA and protein sequences are new boundary conditions, and their construction persistently changes the very phase space of the cell and organism. 

In particular, cells have genetic regulatory networks that turn different genes “on” and off”, so different cell types make different combinations of proteins, each of which is a boundary condition. So different cell types, expressing different proteins that are boundary conditions, “live” in different phase spaces. In single cell and multicell organisms, over the life cycle, the RNA and proteins present change. These changing RNA and proteins sequence contain boundary conditions that are causally effective information that creates different phase spaces.

### The Code Script Is a Set of Changeable Boundary Conditions

The critical step I now take is to identify the “code script” to which Schrödinger points as the very boundary conditions constituted by the aperiodicites in the aperiodic solids in a living cell or system.

A specific example is the hereditary DNA molecule with, in this case, the aperiodic sequence of nucleotide bases, A, T, C and G.

The sequence of bases, A, T, C, and G in the DNA are indeed boundary conditions. DNA is transcribed into RNA, in the simplest case into messenger RNA, consisting of bases A, U, C, and G. That RNA sequence is loaded with twenty different encoded amino acids via specific transfer tRNAs, each of which binds a specific amino acid at its amino acid binding site and binds the RNA sequence’s codon via the transfer tRNA anticodon site. Specific encoded synthetase enzymes attach the proper amino acid to its proper transfer tRNA. As the RNA is so loaded it is processed by the ribosome to link the amino acids into a growing peptide by forming peptide bonds. Work is done at each step utilizing the degradation of ATP to ADP as new bonds are formed.

At all steps, one or more molecules act as boundary conditions on the reaction occurring. For example, the single-stranded DNA molecule uses Watson Crick bonding to bind and line up adjacent free A, U, C, and G nucleotides of the growing RNA molecule. By so binding these free RNA nucleotide bases, their three-dimensional freedom to diffuse is confined to one dimension, thereby lowering the activation barrier to form the proper 3’-5’ phosphodiester bond between adjacent RNA nucleotides. In forming those bonds, thermodynamic work is done, using ATP degradation to ADP as the energy source, and a new chemical bond is formed. 

In short, the single DNA strand acts as a boundary condition “enzyme”, binding the substrate RNA bases, lowering the activation barrier, enabling the chemical energy of the ATP to be released in a few degrees of freedom, thus forming that specific 3’-5’ phosphodiester bond between the two RNA nucleotides rather than the thermodynamically more favorable 2’-5’ bond. 

The DNA, RNA, and encoded proteins carry information because they, counterfactually, are boundary conditions that could have been different. This arises both in evolution by mutation and recombination, and in changing patterns of gene expression in single cells and during development of multi-celled organism with cell differentiation.

In evolution the sequence of bases, A, T, C, and G counterfactually “could have been different”, for an ontologically indeterminate quantum mutation event could have happened and, say, converted an A to a G. This mutation “freely” rearranges the nucleotide sequence in the DNA. 

The “freely” rearranged DNA sequence via the ontologically indeterminate mutation, constitutes a new boundary condition that encodes information because it *is* a boundary condition so causally efficacious and, counterfactually, could have been different had the mutation not occurred by a random quantum event.

I conclude that Schrödinger’s code-script is causally efficacious, and includes the DNA, RNA, and proteins that act efficaciously as freely changeable boundary conditions that both alter the phase space of the system as the boundary conditions change, and by constraining the release of energy, enable thermodynamic work to be done. Other features of cells, such as the cell membrane, organelles, microtubules, are also boundary conditions.

Thus, the code-script is written in a changeable set of boundary conditions that allow thermodynamic work to be done in catalyzed chemical reactions in different specific ways. 

## 13. Work and the Delay in the Production and Release of Entropy

Consider again the cannon and cannon ball. The cannon boundary condition yields the constrained release of energy when the power explodes, which thrusts the cannon ball out the cannon. This is thermodynamic work.

Were the cannon not present, the power explosion would have produced and released entropy *more* rapidly and *less* work would have been done than if the cannon is present, where *more* work is done and the production and release of entropy is *more* delayed. Thus, in general, work is the constrained release of energy into a few degrees of freedom. If the constraints increase so that energy is released into *fewer* degrees of freedom, *more* work is done, and the production and release of entropy is *more* delayed. 

This highly constrained release of energy and the corresponding delay in the production and release in entropy is central to the delay of living cells in falling to thermodynamic equilibrium about which Schrödinger speaks. 

## 14. The Answer to Schrödinger’s Second Question

This is how, in answer to Schrödinger, life copes with the Second Law: Living organisms remain stably organized as non-equilibrium systems despite the Second law. Because life evolved to accomplish highly constrained release of energy into very *few* degrees of freedom in which *more* work done in the cells, the production and release of entropy is *more* delayed so more easily radiated from the cells to the environment. For example, if the production of entropy is delayed in a slowly cooling canon until warm day turns to cold night, more energetic, blue-shifted photons are radiated to cold space and the entropy increase of the planet is less. 

Schrödinger’s aperiodic solids, the answer to his first question, are also the changeable boundary conditions that have evolved to strongly constrain the release of energy in non-equilibrium reactions in cells such that more work is done and production and release of entropy is more delayed; Schrödinger’s answer to his first question answers his second question. 

## 15. Constraint Closure

Automobiles do not construct their own boundary condition. We organize work to construct boundary conditions and assemble them with respect to non-equilibrium processes to make automobiles that do propagating work.

Living cells also accomplish propagating work. Bacteria swim upstream in a glucose gradient. Astonishingly, cells also construct the very boundary conditions by which that work is done, in part, to construct the very same boundary conditions that enable the work to be done by which those same boundary conditions are constructed. 

Mael Montévil and Mateo Mossio introduced the concept of “*Constraint Closure*” in 2015 [8].

In a Constraint Closed system there is a set of non-equilibrium process, 1,2,3… each constrained by a set of constraints, A,B,C… such that the constrained set of processes, 1,2,3…does the thermodynamic work to create the very same set of constraints, A,B,C…

For example, process 1, constrained by A, constructs B. Process 2, constrained by B, constructs C. Process 3, constrained by C, constructs A.

This *constraint closed* system is remarkable. It is an open thermodynamic system carrying out a cycle of thermodynamic work, with the strongly constrained release of energy doing work and with the production and release of entropy delayed. The entire system constructs the very constraints on the release of energy by which the work is done to construct the very same constraints. These constraints closed systems are open and far from equilibrium. Constraint closed systems are “organized” [4,8]. 

But there is more: Living cells are “collectively autocatalytic” [4,9]. No molecule in a collectively autocatalytic set catalyzes its own formation. The catalytic synthesis of each molecule is mediated by other molecules. But the catalysts are themselves the very boundary conditions that the autocatalytic system itself constructs. These same boundary condition catalysts constrain the release of energy doing the work to construct the very same boundary conditions. Thus, a collectively autocatalytic set of molecules also achieves “Constraint Closure”. Such a non-equilibrium-reproducing molecular system literally constructs itself [4].

## 16. A Non-Mystical Holism by Which Cells Construct Themselves and Evolve to Co-Create Biospheres

A non-mystical holistic closure is achieved in the system above [4]. Three Closures are achieved: (1) No constraint constructs itself; the set of non-equilibrium process and constraints mutually do thermodynamic work to construct the joint set of constraints by linking the set of non-equilibrium processes into a closed set of work producing the same set of constraints. This is Constraint Closure. (2) Cells achieve “Catalytic Closure”: Each reaction or non-equilibrium process which must be catalyzed finds a catalyst in the system itself. The catalysts themselves are constraints on the release of energy by which the work is done to construct the other catalysts. Thereby the collectively autocatalytic system also achieves Constraint Closure. (3) A closed set of work “tasks” is accomplished. In the example above the three processes 1, 2, and 3, constrained by A, B, and C are three “work tasks”. All three tasks are accomplished [4]. 

By this non-mystical holistic closure, cells literally construct themselves.

There is, again, more. Such a self-constructing autocatalytic system can construct further constraints on the release of energy in further non-equilibrium processes such that further propagating work occurs. Cells do work to construct the flagellar motor and flagellum by which it moves.

## 17. Schrödinger’s Third Question: Are New Laws of Physics Required?

Living cells evolve by heritable variation and natural selection. In that evolution new DNA, RNA and proteins arise. These may serve as new constraints on the release of energy in new reactions by which work is done. Or the new protein may form a new structure such as a molecular motor.

Over 3.7 billion years, in open-ended evolution, living cells literally co-construct the biosphere in which we all live. Organisms carry with them their history. There is a kind of “propulsive constraint construction”. Constraint closure is always maintained, even as new constraints come into existence. Living lineages branch and diverge in different directions into the “Adjacent Possible” of the biosphere. The “Adjacent Possible” is what can arise next given what is actual now [4,6]. 

These “new directions” into the Adjacent Possible cannot even be pre-stated, let alone predicted. The issue is exemplified by Darwinian “preadaptations”. The swim bladder evolved from the lungs of lung fish. Some water got into the lungs of some fish. The sac was filled with air and water, poised to evolve into a swim bladder, where the ratio of air to water in the bladder assesses neutral buoyancy in the water column. This involved the use of causal features of no prior selective significance in the lung fish that became useful in a new way. Once the swim bladder emerges, a new function, neutral buoyancy, has emerged in the biosphere. But we could not pre-state the emergence of this new function. We do not know what is “in” the Adjacent Possible of the biosphere. Since we do not know the sample space of the process, we cannot have a probability distribution or define “random” [4].

There is even more. Once a swim bladder emerges, a worm or bacterium can evolve to live in the swim bladder. Natural selection “crafted” a functioning swim bladder for selective reasons. But natural selection did *not* craft the swim bladder in order to constitute a new niche for an evolving worm or bacterium.

Stunningly, without selection “achieving it”, the biosphere constructs the very un-prestateable possibilities into which it evolves [4].

The persistent emergence of ever new and un-prestateable functions enables the open-ended evolution of the biosphere [10].

## 18. Answering Schrödinger’s Third Question

Because we cannot pres. tate the new functions that emerge and form the ever-changing phase space of the evolving biosphere, we can write no laws of motion for the evolving biosphere. We cannot integrate the equations we do not have. Therefore, no Newton-like laws entail the evolution of our or any biosphere among the 10 to 22 estimated solar systems in the universe [11].

The un-entailed evolution of life is based on physics, but beyond the entailing laws of physics alone [4,12]. Physics, classical and quantum, is within the Newtonian Paradigm of laws of motion in differential equation form in pre-stated phase spaces, and integration of those laws to yield entailed trajectories in those phase spaces. The evolution of life is based on physics, but entirely beyond the Newtonian Paradigm [4,12]. 

Schrödinger’s third question asked if life required new laws of physics. One wonders what Schrödinger would have thought at the astonishing conclusion that there are no *new* entailing laws of physics, but *no* entailing laws of physics at all for evolving biospheres. Perhaps he would have been pleased. 

## 19. Conclusions

Nobel Laureate Erwin Schrödinger played a major role in bringing modern physics to the considerations of biology. He asked three fundamental questions:
What is source of order in organisms? He answered with the aperiodic solid.How do organisms remain ordered in face of the second law? He supplies no answers. The current article attempts to show that the answer to his first question, aperiodic solids, is the answer to his second question. Aperiodic solids, such as enzymes, can serve as changeable boundary conditions on the constrained release of energy by which thermodynamic work is done. Because the boundary conditions can change, they carry information. More, evolved cells sharply constrain the release of energy to synthesize their products with high efficiency. The more constrained that release of energy is, the more work is done and the more the production of entropy is delayed. The delay in the production of entropy allows that entropic disorder to be rapidly transferred from the organism to the environment. This transfer lowers the increase in entropy of the living system.Schrödinger’s third question asks whether new laws are needed to explain how living cells remain reasonably ordered in face of the Second Law of Thermodynamics. Our results suggest that, with respect to holding entropy low in living cells, no new laws of physics are required. This is one answer to Schrödinger’s third question.

But more broadly, we can ask if the evolution of life requires new laws of physics. Here, the answer is remarkable. Life is based on physics but no laws at all entail the evolution of our or any biosphere. Classical and Quantum Mechanics are within the Newtonian Paradigm. The evolution of life is entirely beyond the Newtonian Paradigm. Three hundred and thirty years after Newton and his entailing laws, we learn that our biosphere, the most complex system we know in the universe, literally constructs itself in a self-consistent manner into the growing Adjacent Possible evolution un-prestataby creates for itself. The evolution of life is astonishing.

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
