# Peer review of "Answering Schrödinger’s “What Is Life?”"

_entropy, 2020, doi:10.3390/e22080815_

Round 1
Reviewer 1 Report
This paper presents many of the key themes that have been developed in Kauffman's work over recent decades. For those who are familiar with Kauffman’s work, this is a relatively simple presentation and may represent a helpful entry to those not familiar. The one point that I, at least, had not seen emphasized in earlier work was the slowing of the increase in entropy. It is also not clear how important that is to showing how Kauffman’s account shows how aperiodic structures serve to answer Schrödinger’s second question.
Basically, I think this paper is acceptable. I have inserted a few more substantive comments below. But the writing needs extensive work. I have identified some of the problems.
- 78: no close quote
- 90: “use of” should be deleted as well as the quote marks around “sucking . . . environment.” It is the phenomenon that is also called . . .
- 114: replace “therefore” with “thereby”
- 127: replace “, by” with “that results from”
- 134-5: something is seriously wrong here. Maybe delete “that” in l. 134 and “is” in line 135. I would also recommend dividing the sentence into two.
- 134: what is the “(ibid)” doing?
- 135: delete “in their complexity”
- 146-150: make into two sentences
- 156: delete comma and put period at end.
- 159: I am not sure what the “i.e., Newton” is doing. Period needed at end.
- 160: need is after second “work”
- 161: delete first Atkins and period
- 163: role of phrase “as created by the cannon itself.” Maybe just “by the cannon.”
- 186: “process” should be “processes”
- 187: why the capitalization of “Propagating Work”
- 198: replace “what” with “was
- 212: add comma after pistons
- 213: make “condition” plural and add “or”
- 215: something is missing. Maybe the intent is “processes that allow our”
- 225: “framework” is singular.
- 226 ff.: this is hard to parse. Maybe start with “motion that couple”
- 229: at least add a hyphen in “differential equation”. Or maybe plural equations and then “that embody the laws of motion”
- 234: I am not sure how the boundary conditions “thereby specify all possible positions”
- 234: why the “But”?
- 249: add “and” between “long” and “short”
- 251: no comma after “hit”
- 261: repeats last sentence of previous paragraph.
- 264: it is the existence, not the synthesis, that constitute the boundary conditions. Divide this sentence into two or more.
- 274: “so”? why does the network make for different combinations?
- 277: doesn’t the cell cycle apply as well to cells in multicellular organisms that are still dividing. Even when they are not, the RNA and proteins change and change boundary conditions.
- 288: after “RNA,” add “consisting of the bases”
- 289: not sure what it means to say that an “RNA sequence is loaded.” Loaded into the ribosome?
- 295: might help to make it explicit that the ATP provides the energy required for the work to be done. This is made clear on l. 303 but by then the sentence seems repetitious.
- 306 ff: no clear that these level of detail adds anything to the account.
- 317: might help to unpack “could have been different” a bit—could be different and not change the laws governing the system—only the constraints operative.
- 340: powder? And “which” before “thrusts.”
- 349: What is with the question mark?
- 358: add “is” before “done”. What is the import of “in the cells”? I think it is highly significant since cells are the sorts of things that deploy chemical reactions to make boundary conditions, but this has not been made clear.
- 359: isn’t it the radiation that is delayed?
- 360: need articles before “warm day” and “cold night”
- 362: not clear. If more photons are released, won’t entropy increase faster? In any case, the relevance of this point is not clear.
- 378: delete “by which . . .”
- 381: I would have thought that the closure of constraints should have appeared earlier—it explains why cells are the relevant unit.
- 392-4: this sentence is hard to parse
- 395: What is the significance of “organized”—note, Moreno and colleagues use it to characterize their approach.
- 408ff. From the previous discussion, constraint closure is relatively clear. But not the other two. How do the three notions of constraint relate?
- 414: hard to parts the phrase “in-opened evolution”
- 420: Kauffman ends with a dangling question about laws. Part of it concerns what to count as a law: just the statements of relations without constraints, or those in which constraints are included? Given the open-endedness of constraints, the latter position leads to a incredible increase in the number of laws—the differential equations describing any given reaction, no matter how specific the enzymes, will count as laws. Not clear how Kauffman comes down on the use of the term “law”.
Author Response
Thank you for your thoughtful and thorough review.

Reviewer 2 Report
This paper is focused on a comparison between Schrödinger' s principle of "order by order" formulated in "What is life?", and the "constraint-work cycle" proposed by Stuart Kauffman in many books, since Investigations (2000).
However, this paper also proposes a description of the "constraint-work cycle", that could be linked with the principle of "constraint closure" initially proposed by Mossio/Moreno (2010), and developed in a very interesting paper, by Montévil/Mossio (2015).
1- My first question: the principle of "constraint-work cycle" means also that work propagates, so that the more energy is released, the more new constraints can emerge. It doesn't mean simply that the work and constraint cycle is closed on itself, in such a way that the same constraints are continuously regenerated by the same processes and by the same set of constraints. How do you deal with this important difference?
Organizational closure, as presented in 2015, can explain how the organism can maintain itself, but it cannot explain how it can change during time. Thus, it is important to specify if organisms realize closure with the same constraints, or not ! If the "constraint-work cycle" can be characterized as a virtuous cycle, it can be closed on the fact that it produces new constraints, during time.
2- Montévil and Mossio know that very well, and that is why they propose a very different diagram in their last published paper (Frontiers of physiology, 2020). In this new diagram the past of a biological system, is not simply some kind of initial or boundary condition. It is some kind of historical constraint (KI) integrated in the set of constraints characterizing organizational closure. It explains immediately why new "unpredictable variations" could appear. It also explain the presence of propulsive constraints in a biological system. In my opinion, this concept of propulsive constraint could be directly connected with Kauffman's concept of propagation. What do you think of this?
3- In your last book (2019) you explain very well, that there is not simply one, but three closures, so that a biological system could have a metabolism that feeds it and a catalytic task closure, through which it can reproduce itself. This very interesting idea is not explicitly developed in this paper. It seems to me that the beginning of the paper is too long, and that the end of the paper is written too quickly. The connection with an open ended evolution has also to be more explicit.
4- It could be also interesting to compare your vision of a self-reproducing system characterized as an autocatalytic set, and the concept of "kinetic dynamic stability" proposed by Pross and Pascal (J.O. Chemistry, 2017). They are not focused on biochemical reactions, and they assume that natural selection can emerge in chemical systems, that are intrinsically heterogeneous.
Author Response
In response to reviewer 2, I have enlarged the discussion in the final quarter of the paper. Thank you for your thoughtful review. Please see the attachment.

Round 2
Reviewer 1 Report
Here are a few areas requiring edits
l. 110 and following: cannot parse
l. 160: change "what" to "was"
l. 186: delete "thereby"
l. 208: Edit "In evolving organisms, with DNA RNA and encoded protein synthesis, once synthesized, these molecules are" to "In evolving organisms, DNA, RNA, and proteins, once synthesized, are"
Reviewer 2 Report
I am pleased with the corrections.